psychology

maths attainment, school transition, parent–child relationships, parental support, Avon Longitudinal Study of Parents and Children

**Author for correspondence:**
Danielle Evans
e-mail: de84@sussex.ac.uk

# Predictors of mathematical attainment trajectories across the primary-to-secondary education transition: parental factors and the home environment

## Danielle Evans and Andy P. Field

School of Psychology, University of Sussex, Brighton, UK

DE, 0000-0002-5330-3393; APF, 0000-0003-3306-4695

A 'maths crisis' has been identified in the UK, with many adults and adolescents underachieving in maths and numeracy. This poor performance is likely to develop from deficits in maths already present in childhood. Potential predictors of maths attainment trajectories throughout childhood and adolescence relate to the home environment and aspects of parenting including parent–child relationships, parental mental health, school involvement, home teaching, parental education and gendered play at home. This study examined the aforementioned factors as predictors of children's maths attainment trajectories (age 7–16) across the challenging transition to secondary education. A secondary longitudinal analysis of the Avon Longitudinal Study of Parents and Children found support for parental education qualifications, a harmonious parent–child relationship and school involvement at age 11 as substantial predictors of maths attainment trajectories across the transition to secondary education. These findings highlight the importance of parental involvement for maths attainment throughout primary and secondary education.

## 1. Introduction

Carey *et al.* [1] propose a 'maths crisis' in the UK, with a staggeringly high proportion of adults underperforming in maths and numeracy. Recent statistics show that around half of working-age adults in the UK have maths skills no better than 6-year-old children, and only 22% have the skills expected of an 'average' 16-year-old—a 4% decrease from 8 years prior;

suggesting that the problem is getting worse [2]. In support of this decline, figures from the Programme for International Student Assessment (PISA) show that the performance of the UK has not improved since 2012 [3,4]. Low levels of mathematical and numerical skills are detrimental to individuals and to wider society. For individuals, low maths attainment is associated with greater unemployment and poor career prospects, increased mental and physical health issues, a higher likelihood of homelessness and lower socio-economic status (SES) [5–8]. At societal levels, poor numeracy is reported to cost £2.4 billion a year through expenses such as unemployment benefits, lost tax revenue and increased contact with the criminal justice system [9].

The deficits in maths and numeracy seen in adults in the UK, are likely to stem from difficulties with maths in childhood. Therefore, understanding the factors that predict maths attainment in childhood is important to inform educational practices. In response to the paucity of research investigating long-term influences on maths attainment, this study aims to identify early predictors of maths attainment in childhood and adolescence, focusing specifically on both the impact of parents and the challenging transition from primary to secondary education.

## 1.1. How do parents contribute to maths cognition and learning?

Children's first educators are their parents. The early experiences parents provide for their children lay the foundations for what follows in formal schooling by developing skills and promoting the desire for the acquisition of knowledge. Parents play an extremely important role within their child's educational success with both positive and negative effects. We know that adversities present as early as pregnancy and negative experiences within the first few months of life can affect the development of cognitive abilities, negatively affecting the trajectory of educational attainment long-term. For example, smoking and drug misuse during pregnancy, other adverse childhood experiences (ACEs; i.e. domestic violence, child abuse and neglect), and poor parental mental health and well-being are linked to decreased cognitive development and academic abilities [10–15].

There is also the opportunity for positive experiences for growth and development in childhood that are provided by parents. Greater participation in educational activities at home and in school is related to increased cognitive abilities and greater educational motivation, engagement and success [16–19]. Parents also contribute to developing early maths skills, the transmission of attitudes to, interest in, and the value given to maths, and influence their child's involvement in educational activities (at home and in school). One of the strongest predictors of maths attainment in later childhood are the very early skills children have when starting school [20–23]. Parents who provide greater opportunities for teaching maths and numbers at home significantly increase their child's future maths skills and achievement [24–28]. Moreover, early numeracy teaching is not *only* associated with maths outcomes, but is also correlated with increased vocabulary, more so than literacy activities [29].

The contribution of parents to their child's maths attainment does not cease with the start of school. Several researchers report parental influence on aspects of maths throughout childhood and adolescence. Cai [30] found that greater parental involvement in primary education was associated with higher school grades. Moreover, parents have been found to transmit maths anxiety, attitudes towards and interest in maths [31–34], all of which are associated with maths attainment [35–37].

Parents additionally guide their children's interests and the types of play they participate in by purchasing toys and encouraging (or discouraging) different types of activities and behaviours. One specific area of interest particularly for maths attainment is gender-stereotyped play in childhood. Typically, 'boy toys' include construction toys (such as building blocks and tools), vehicles and sports, and 'girl toys' include dolls, household toys (i.e. tea sets and toy kitchens) and 'dress-up' [38]. This divide is particularly interesting given reported differences in maths attainment between males and females [4], which could potentially stem from the differences in toys and play through the increased spatial content in 'masculine' toys for example. Because parents heavily shape their child's preferences and activities, examining gender-stereotyped play in childhood could help understand differences in attainment for males and females stemming from parents and aspects of the home environment.

It is evident that parents play a pivotal role within achievement which is particularly relevant in early and middle childhood [39]. Adolescence presents additional challenges that affect the acquisition of maths skills generally. Research suggests that the shift from childhood to adolescence can be a critical period for several reasons, some of which relate to the effects associated with the transition to secondary education, as now discussed.

## 1.2. The transition from primary to secondary education

The transition to secondary education usually occurs between ages 10 and 14, where young adolescents move from a typically small primary school, to a much larger secondary education institution. The transition is the norm for most students in Western societies (e.g. UK, Europe, USA, Australia and New Zealand), and is regarded as one of the most stress-inducing events young adolescents will encounter within their development and education [40,41]. The transition to secondary education elicits several changes within a child's social and educational environment. Secondary education schools are typically much larger than primary schools, with several specialized subject teachers (compared with just one teacher per school year in primary education). The transition usually involves the loss and renegotiation of friendships [42]. Relationships with parents also change with the transition, with parents granting more autonomy and independence to their children as they grow into adolescents. Moreover, the transition occurs alongside the onset of puberty, and so coincides with further biological, emotional and social changes.

The transition can have negative psychological consequences [43], including increased anxiety [44], relationship concerns [45], increased loneliness [44], fear of victimization [45] and fear of being lost or late for class [45,46]. Individuals reporting greater concerns regarding the transition experience increased anxiety both pre- and post-transition also [47]. Many of the concerns reported by parents and children are related to practical or relational issues (i.e. making friends and getting to class on time [46]). However, research shows that academic achievement is also negatively impacted by the transition with declines in achievement and a lack of progress made across the transitional year [48–50].

### 1.2.1. Effects of the education transition on maths outcomes

The transition to secondary education influences the acquisition and performance of maths skills and abilities. It has been reported that 34% of children fail to make any progress in maths during the transition year [51], while other researchers have found that maths anxiety increases [52]. Furthermore, enjoyment and interest in maths decreases across the transition, with children becoming less involved in maths class [53], and their attitudes towards maths become more negative in secondary education [53,54]. Additionally, poor maths attainment across the transition predicts greater maths anxiety at age 18 [55], suggesting that this is a critical period to focus on when aiming to improve maths education outcomes. However, additional research within this area is needed to further understand the mechanisms contributing to low maths attainment.

### 1.2.2. Parental effects and the transition to secondary education

As previously highlighted, the transition to secondary education is a turbulent time associated with negative emotional and maths outcomes. A well-adapted transition is dependent on several interacting risk and protecting factors [43], with some of the most relevant factors relating to parent–child interactions, familial support and the home environment.

Parental support is reportedly one of the most important support systems children have when transitioning to secondary education [56–58], and can affect the success of the transition process in several ways. For example, greater parental presence at home has been found to be protective against a difficult transition [59]. Moreover, low parental support is strongly related to greater emotional problems experienced by adolescents, with high support being an especially protective factor found during early adolescence [56], though generally, perceived parental support is found to decline significantly during this period [56]. Children's attachment to their mother also predicts worries about the transition (related to academic and teacher domains), through anxiety symptoms, meaning that a secure mother–child attachment is protective against intrusive concerns and anxieties surrounding the transition [60]. Moreover, increased autonomy support provided by parents (i.e. greater independence) prior to the transition predicted a decline in depression post-transition [61]. Though many of these aspects relate to 'emotional support', previous studies have found that children with fewer family 'resources' (such as low SES and parental education), have worse academic [50] and maths [62] performance across the transition, and that this may be explained by decreased parental support [50]. However, it is currently unknown how these parental factors might affect maths attainment trajectories across the transition.

## 1.3. The present study

To summarize, it is apparent that parents are influential for the development of maths skills and play an important role for a successful transition to secondary education. Moreover, the transition itself is associated with negative maths outcomes, raising the question of how parental factors might affect maths attainment trajectories across the transition to secondary education. Understanding which factors in childhood have negative impacts on the trajectory of maths attainment will help enable strategies for mitigating the difficulties many adults have with mathematics. There is a paucity of longitudinal research investigating maths attainment across the transition to secondary education, which as discussed previously, is a critical time for maths interest and attitudes, maths achievement and maths anxiety. Given that parents themselves are *at least* partly involved in the development of maths skills and the transmission of maths attitudes generally, and that parental support can buffer negative effects surrounding the primary–secondary education transition, it seems logical to investigate how parents might affect maths attainment across this transition.

Therefore, by using secondary analysis of the Avon Longitudinal Study of Parents and Children (ALSPAC), this study aims to investigate parental influences in childhood and early adolescence as predictors of maths attainment trajectories for typically developing children across the transition from primary to secondary education. ALSPAC is a large UK birth cohort following children and their parents from pregnancy up to the present day, and covers a broad range of measures. Previous work investigating home and parental factors using the ALSPAC cohort have focused particularly on the impact of mothers and have shown that that maths attainment is associated with maternal perinatal and postnatal mental health [63,64], and maternal prenatal locus of control [65]. Parental education qualifications at birth are also linked to maths attainment in the ALSPAC sample [62], though, mothers' participation in adult learning was not found to improve maths grades [66]. The current study aims to add to this existing literature by focusing on the influence of the home environment and parental factors on the trajectory of children's maths attainment (measured from age 7 up to age 16 using national curriculum assessments). This study looks at several indicators of the home environment and parental factors including: early home teaching, parental mental health, parent–child home interactions, parent–child relationships, school involvement, parental education qualifications and child gendered play. The analysis includes contextual variables (socio-economic status, child IQ and biological sex) and measures of working memory and internalizing symptoms which were found to predict maths attainment in this sample by a previous study [62].

Based on existing research, it is predicted that greater involvement (i.e. home interactions, home teaching and in school), positive parent–child relationships, greater parental education qualifications, and male-gendered play will be positively associated with maths attainment at the transition, and the growth in attainment over time. It is predicted that increased parental mental health issues will negatively impact attainment at the transition and predict a decreased rate of growth over time.

# 2. Method

## 2.1. Sample

This paper describes a secondary analysis of data from the Avon Longitudinal Study of Parents and Children (ALSPAC). ALSPAC is a large birth cohort consisting of children born to women residing in the southwest of England with a due date between 1 April 1991 and 31 December 1992 [67,68]. The children and their parents have been followed up extensively from pregnancy through to the present day. The core ALSPAC sample recruited initially consisted of 14 062 live births, of which 13 988 children were alive at 1 year. Additional participants were recruited resulting in a total of 15 589 fetuses, of which 14 901 were alive at 1 year. There is a slight over-representation of white families with higher socio-economic status [67], but generally, the sample is representative of the overall population.

The data consisted of self-report postal questionnaires, completed by the study child, the child's mother/father/carer, the child's teacher(s) and the mother's partner. This study also uses education-linked data from the National Pupil Database (NPD). Some of the data were collected through 'Children in Focus' clinics, which were attended by a smaller subsample (10%) of participants. Analysis was conducted on singletons and the first-born twin. The final sample size was 7465 (see Exclusions and missing data section for details).

The ALSPAC website has a fully searchable data dictionary and variable search tool (see http://www.bristol.ac.uk/alspac/researchers/our-data/). All participants provided written informed consent

prior to the study. Ethical approval was obtained from the ALSPAC Ethics and Law Committee and the Local Research Ethics Committees. Informed consent for the use of data collected via questionnaires and clinics was obtained from participants following the recommendations of the ALSPAC Ethics and Law Committee at the time.

## 2.2. Outcome

### 2.2.1. Maths attainment

In England, children in formal education are assessed at 'key stages' up until age 16. There are four key stages relating to different phases of development with exams or assessments at the end of each stage to evaluate the child's progress. For this study, maths grades at age 6–7 (key stage 1), age 10–11 (key stage 2), age 13–14 (key stage 3) and age 15–16 (key stage 4) were obtained from external education records (National Pupil Database). In key stages 1–3, children are assigned a numerical grade based on their performance ranging from 1 to 8 with a higher score indicating a better grade. The grades children are expected to attain at each of these key stages is as follows: level 2 at key stage 1, level 4 at key stage 2 and levels 5–6 at key stage 3. At key stage 4, adolescents can achieve an alphabetical grade from the highest of 'A*', through 'A', 'B', 'C', 'D', 'E', 'F', 'G' and the lowest grade of a 'U', which were coded into numbers between 2 and 10, with 10 being the highest grade achievable (i.e. A*). Maths attainment at age 10–11 (key stage 2) just prior to the transition to secondary education, and the growth in attainment over time were the main outcomes of this study.

## 2.3. Predictors: parent and child

It is important to highlight that measures regarding the child's 'parent(s)', refer to the child's primary carer. Data on the specific demographics are not available for this dataset, but from the entire ALSPAC sample, this mostly consists of the child's biological mother and father, and also includes non-biological parents (i.e. step/adoptive), grandparents and other legal guardians. For simplicity, the term 'parent' is used throughout, and refers to any of the aforementioned carers, unless otherwise specified.

### 2.3.1. Postnatal parental mental health

Parental mental health was assessed using the Crown-Crisp Experiential Index (CCEI; [69]) total score, and the Edinburgh postnatal depression (EPDS; [70]) score. The CCEI used in ALSPAC includes 23 items relating to somatic, depressive and anxious symptoms with a possible score ranging between 0 and 46, with a higher score indicating more symptoms. The EPDS is mostly used to assess postnatal depression but can also be used to indicate depression in those that have not recently given birth [71]. Possible scores on the EPDS range from 0 to 30 with a higher score indicative of greater symptoms. The CCEI was administered at eight weeks post-birth and 21 months post-birth, and the EPDS was administered at eight weeks post-birth.

The highest available score for either parent was taken from the CCEI at time 1, time 2 and the EPDS separately, meaning that the score could be from either parent for the three measures. An exploratory factor analysis (using parallel analysis) revealed one common factor (Cronbach's $\alpha = 0.81$), so a composite score was created by extracting factor scores, calculated using the regression method.

### 2.3.2. Parental interactions at home

Parent–child home interactions were measured at age 3.5 years and age 6.75 years. At age 3.5, 'mothers' (i.e. the primary female carer) were asked to indicate whether they, and their partner, participated in the following activities: 'bathes child', 'feeds child', 'sings to child', 'reads stories/shows pictures', 'plays with child with toys', 'cuddles child', 'plays imitation games (peek-a-boo)', 'physically plays with child' and 'takes child for walks'. Responses included *never*, *less than once a week*, *one to two times a week*, *three to five times a week* and *daily* (coded as 0–4).

At age 6.75, mothers were asked if they, and their partner, participated in the following activities: 'bathes child', 'makes things with child', 'sings to child', 'reads to or with child', 'plays with child with toys', 'cuddles child', 'does active play (ball games, hide and seek)', 'takes child to park/playground', 'puts child to bed', 'takes child swimming', 'fishing, or similar activity', 'paints or draws with child', 'prepares food for child', 'takes child to classes', 'takes child shopping', 'takes child to

watch sports/football', 'does homework with child', 'has conversations with child', 'helps child prepare stuff for school' and 'does other activity with child'. Responses included *never, less than once a week, once a week, two to five times a week* and *daily*, scored from 0 (*never*) to 4 (*daily*).

The two measures had a different number of items so the score for the second timepoint was transformed to be on the same scale as the first timepoint. Factor analysis revealed two factors (determined by parallel analysis) relating to mother's interactions and partner's interactions separately, so the scores for each parent were then averaged to get a composite score for mother's interaction (for time 1 and time 2) and partner's interaction (for time 1 and time 2). Demographic data for the mother's partner are not available from this specific dataset; however, broadly across the entire ALSPAC dataset, very few mothers had female partners (time 1: 0.2%; time 2: 0.1%), relatively few were without a partner, or not living with a partner (time 1: 7%; time 2: 10.8%), and the majority had male partners (time 1: 91.9%; time 2: 86.3%). Possible scores ranged from 0 to 36 with a higher score equating to a greater parent–child interaction.

### 2.3.3. Early home teaching

Parents were asked to indicate if they taught their child the alphabet, colours, numbers and shapes at 18 months old. Polychoric factor analysis (using parallel analysis) revealed two factors which were split into verbal and numerical skills which are referred to as 'literacy' (teaching alphabet and colours), and 'numeracy' (teaching numbers and shapes) home teaching throughout. Possible scores for each measure could range from 0 (not taught any skills) to 2 (taught both alphabet and colours, *or* numbers and shapes).

### 2.3.4. Parental education

Parental education was measured by asking parents their highest qualification at 32 weeks gestation. This response was coded into the following five categories: 'no qualifications/no higher than CSE or GCSE', 'vocational qualifications (i.e. teaching or nursing qualifications)', 'O-level or equivalent', 'A-level or equivalent' and 'university degree'. The highest qualification of either parent was used in analysis. Frequencies for each category are as follows: 8.0% had a CSE or below, 5.3% had a vocational qualification, 26.1% had an O level, 34.8% had an A level and 25.9% had a degree. Having the highest qualification of a CSE (or below) was used as the reference group as this was the lowest level of parental education qualifications available.

### 2.3.5. Gender-stereotyped play and behaviour

Gendered play was included as an indicator of the types of toys parents provided and the play they participated in with their children at home. The Pre-School Activities Inventory (PSAI; Golombok & Rust [72]) was used to examine the child's gender-stereotyped play. At 3.5 years, parents were asked to state how often their child plays with the following: 'guns (or similar)', 'jewellery', 'tool set', 'dolls', 'cars/planes/trains', 'swords (or similar)', 'tea set', 'played house (i.e. cleaning/cooking)', 'played with girls', 'pretended to be a female character (i.e. a princess)', 'pretended to be a male character (i.e. a soldier)', 'played fighting', 'played at being a mother/father', 'ball games', 'climbed (tree, fence, climbing frame)', 'played at looking after babies', 'showed interest in real cars/planes/trains', 'dressed up in female clothing', 'explored new surroundings', 'rough and tumble play', 'showed interest in insects/spiders/snakes', 'avoided getting dirty' and 'avoided taking risks'. Parents responded on a 5-point scale: *never, hardly ever, sometimes, often* and *very often* (scored from 1 to 5). The items are then scored in a way that a high score on the PSAI indicates more 'masculine' behaviour.

At 8 years old, children participated in the Children's Activities Inventory (CAI; [72,73]), which is a shorter version of the PSAI for older children. Children were asked to indicate how much agreement they have when asked if they play with the following toys: 'jewellery', 'computer games', 'dolls', 'tea sets', 'guns', 'house (cooking and cleaning)', 'playing with boys', 'pretending to be a female character', 'fighting', 'sports', 'climbing', 'taking care of babies', 'dressing up in female clothing', 'being outdoors', 'wrestling' and 'liking pretty things', on a scale of *yes, definitely like me*; *yes, a bit like me*; *no, not really like me* and *no, not at all like me* (scored from 1 to 4). As with the PSAI, the individual items were transformed so that a high score indicated more 'masculine' behaviour.

### 2.3.6. Parental school involvement

Parental involvement with school activities was assessed at age 8 and age 11. The child's school teacher was asked to indicate whether the child's parent(s) had been involved in the following activities by

answering *yes* (coded as 1) or *no* (coded as 0) to: 'helping in class', 'helping with out of class activities', 'attending parent-teacher sessions' and 'being involved in another school activity'. Polychoric factor analysis and parallel analysis revealed two factors for age 8 school involvement and age 11 school involvement separately (Cronbach's $\alpha$ = 0.61 and 0.62, respectively). The items were summed to create a score between 0 and 4 for school involvement at age 8 and age 11, with a higher score indicative of participation in more activities.

### 2.3.7. Parent–child relationships

The quality of parent–child relationships at 12.5 years was assessed using the Assessment of Mother–Child-Interaction with the Etch-a-Sketch (AMCIES; [74,75]). This task involves the observation of a parent–child dyad during a play situation with an Etch-a-Sketch toy. Parent–child pairs were asked to draw a picture of a house, with one individual responsible for drawing horizontal lines, and the other responsible for the vertical lines—meaning they have to work closely together in cooperation.

The ALSPAC team rated the observed interaction for 'harmony' and 'control'. Harmony was evaluated by assessing the amount of conflict within the interaction on a 5-point scale: 'many conflicts', 'some conflicts (generally negative with some conflict)', 'neutral (atmosphere is neither positive or negative)', 'quite agreeable (generally positive)' and 'very agreeable (very positive and harmonious)'. Scores can range from 0 to 4, with a higher score referring to greater harmony. Control was rated by assessing whether the child or adult was in control of the session itself and who was determining the outcome of the interaction. Control was rated on a 5-point scale: 'child is in complete control (initiates and/or directs adult)', 'child has most control', 'equal control', 'adult has most control' and 'adult is in complete control (i.e. initiates and/or directs child)'. This variable was recoded so that a high score indicated the child had the highest control (on a scale from 0 to 4).

## 2.4. Predictors: contextual

### 2.4.1. Biological sex

Biological sex at birth was added in as a predictor due to potential sex differences in maths attainment. This variable was dummy coded with females being the reference group. Females accounted for 51.9% of the sample.

### 2.4.2. Socio-economic status

Socio-economic status was measured using the Cambridge Social Interaction and Stratification Scale (CAMSIS). The CAMSIS measures occupational structure based upon social interactions [76] with possible scores ranging between 1 (least advantaged) and 99 (most advantaged), with a mean of 50 and a standard deviation of 15 in the national population [77]. To retain as much data as possible, the CAMSIS score at 32 weeks gestation was used, with the highest score taken from either parent.

### 2.4.3. Internalizing symptoms

The strengths and difficulties questionnaire (SDQ; [78]) was used to measure internalizing symptoms experienced by the child at age 11. The SDQ assesses emotional symptoms, conduct problems, peer problems, prosocial behaviour and hyperactivity. The sum of emotional symptoms and peer problems (10 items in total) was used as an 'internalizing symptoms' score. Example statements of these scales include 'I am often unhappy' and 'I am usually on my own'. Parents rated their child's behaviour on a 3-point scale of *not true* (coded as 0), *somewhat true* (coded as 1) and *certainly true* (coded as 2). The internalizing symptoms score ranges from 0 to 20 with a higher score indicating more symptoms. The SDQ has good concurrent and predictive validity [78], and satisfactory internal consistency (Cronbach's $\alpha$ for emotional difficulties = 0.66, and for peer problems $\alpha$ = 0.53; [79]).

### 2.4.4. Working memory and general intelligence

IQ at age 8 was measured using the Wechsler Intelligence Scale for Children (WISC-III; [80]), as part of one of the Children in Focus sessions. The IQ measure includes short-form tasks assessing five verbal subtests (information, similarities, arithmetic, vocabulary and comprehension) and five performance subtests (picture completion, coding (full-form test administered), picture arrangement, block design and object

assembly). The scorings for the short-form subtests were transformed to be as though the full-form version of the tests had been administered. The test-retest reliability of the WISC-III is good (0.80–0.89; [81]).

Working memory was measured using the counting span task [82] in the age 10 Children in Focus session. Displays of red- and blue-coloured dots were presented on a computer screen, and children were asked to count the number of red dots out loud, followed by recalling the number of red dots in the order they were presented for multiple screens. Children were shown two practice screens followed by three sets of two screens, three sets of three screens, three sets of four screens and three sets of five screens. Children were asked to complete 42 trials in total, reflecting the maximum score children could attain for this measure (scored from 0 to 42).

## 2.5. Data analysis

### 2.5.1. Exclusions and missing data

The initial sample was 14 901 (see Sample section). Withdrawal from the study led to a sample size of 14 684. Of this, data from singletons and the first-born twin were retained for analysis ($N = 14\,498$). Fourteen children were excluded as their first, or second main language was not English ($N = 14\,484$). A further 2652 children identified by teachers (at ages 7–8 and 10–11) as having or having had special educational needs (such as learning difficulties, emotional and behavioural difficulties, physical disabilities, and speech and language difficulties) were excluded ($N = 11\,832$) due to the high heterogeneity within this group. Finally, 4367 participants lacking data for 50% or more of the predictor variables were excluded, leaving a final sample size of 7465 (none of which were complete cases when including outcome variables).

To address the issue of high attrition rates and missing data (for missing data per variable, see table 1), multiple imputation was performed in R [83] using the semTools [84] and Amelia packages [85]. Eighty imputations were performed and the results were pooled [86]. The outcome variables (maths attainment KS1–KS4) were included in the imputation model but were not imputed. Instead, to address the missing outcome data, full information maximum-likelihood (FIML) estimation was used, which has been shown to be a superior method when dealing with missing data [87].

### 2.5.2. Statistical analysis

All analyses were conducted using R v. 3.4.3 [83] and the following packages: lavaan [88], psych [89], polycor [90], nFactors [91], tidyverse [92], mice [93], semTools [84] and Amelia [85].

The lavaan package [88] was used to fit a latent growth model predicting maths attainment trajectories across the transition to secondary education (figure 1). The unconditional growth model was the same as Evans *et al.* [62, p. 7], which they described as: 'maths attainment at 7, 11, 14 and 16 years were endogenous observed variables predicted from latent variables representing the intercept and slope for growth in maths attainment over time. The loadings for the paths from the slope latent variable to the four maths attainment outcomes were constrained to be −4, 0, 3 and 5 so that the intercept represented maths attainment at 11 years old (i.e. the time of the school transition).' The predictors were included as exogenous observed variables that predict the intercept and slope (i.e. the rate of change) of growth in maths attainment.

Scores for SES, IQ, working memory and gendered play (CAI & PSAI) were centred so that the effects would be expressed at average levels of these predictors as opposed to them being zero. A previous study [62] found working memory and internalizing symptoms predicted maths attainment in this sample, so these were included as predictors to adjust for their effects. All predictor variables were entered into the model simultaneously.

# 3. Results

## 3.1. Descriptive statistics and model fit

Descriptive statistics for the model predictors and outcomes are presented in table 1. Table 2 shows a matrix of the correlation coefficients for the numeric predictors and outcome variables. The latent growth model provided satisfactory fit indices, CFI = 0.939, TLI = 0.881, RMSEA = 0.092 [90% CI = 0.089, 0.095], SRMR = 0.043.

**Table 1.** Summary statistics for the key study measures.

| measure | N | min | max | *Mdn* | *M* | 95% CI | *s* | % MD |
|---|---|---|---|---|---|---|---|---|
| SDQ | 5664 | 0.00 | 17.00 | 2.00 | 2.39 | [2.32, 2.46] | 6.52 | 24% |
| IQ | 5651 | 49.00 | 151.00 | 106.00 | 106.44 | [106.04, 106.85] | 240.39 | 24% |
| WM | 5336 | 0.00 | 42.00 | 19.00 | 19.19 | [18.99, 19.40] | 58.03 | 29% |
| SES | 6165 | 23.72 | 95.70 | 58.40 | 59.17 | [58.88, 59.46] | 136.64 | 17% |
| parental MH | 6917 | −1.52 | 4.64 | −0.24 | −0.04 | [−0.06, −0.02] | 0.84 | 7% |
| interaction (M) | 6358 | 9.81 | 31.56 | 23.56 | 23.36 | [23.28, 23.44] | 10.59 | 15% |
| interaction (P) | 5995 | 0.38 | 31.00 | 17.56 | 17.27 | [17.16, 17.38] | 19.37 | 20% |
| CAI | 5460 | 3.45 | 81.35 | 50.95 | 49.43 | [49.02, 49.85] | 244.29 | 27% |
| PSAI | 7050 | 4.25 | 95.55 | 49.35 | 49.16 | [48.80, 49.52] | 240.48 | 6% |
| home teaching (L) | 7263 | 0.00 | 2.00 | 1.00 | 0.99 | [0.97, 1.00] | 0.54 | 3% |
| home teaching (N) | 7263 | 0.00 | 2.00 | 1.00 | 1.33 | [1.31, 1.35] | 0.55 | 3% |
| SI (8) | 3429 | 0.00 | 4.00 | 1.00 | 1.85 | [1.82, 1.89] | 1.18 | 54% |
| SI (11) | 3943 | 0.00 | 4.00 | 1.00 | 1.75 | [1.72, 1.78] | 1.13 | 47% |
| parent–child (H) | 5167 | 0.00 | 4.00 | 3.00 | 3.24 | [3.22, 3.26] | 0.64 | 31% |
| parent–child (C) | 5176 | 0.00 | 4.00 | 2.00 | 1.95 | [1.92, 1.98] | 1.28 | 31% |
| KS1 Maths (7) | 5581 | 0.00 | 3.00 | 2.00 | 2.29 | [2.28, 2.31] | 0.29 | 25% |
| KS2 Maths (11) | 6110 | 1.00 | 6.00 | 4.00 | 4.33 | [4.32, 4.35] | 0.48 | 18% |
| KS3 Maths (14) | 5275 | 1.00 | 8.00 | 6.00 | 6.25 | [6.21, 6.28] | 1.37 | 29% |
| KS4 Maths (16) | 5708 | 2.00 | 10.00 | 7.00 | 7.37 | [7.33, 7.41] | 2.46 | 24% |

Note: Child's age in years are given in brackets for duplicate measures. SDQ, internalizing symptoms; WM, working memory; MH, mental health (factor scores); interaction M and P, mother's home interaction and partner's home interaction; CAI, gendered play (8 years); PSAI, gendered play (3.5 years); home teaching N and L, numeracy and literacy; SI, school involvement; parent–child H and C, harmony and control; KS, key stage; MD, percentage of missing data per variable.

## 3.2. Predictors of maths attainment at age 11 (intercept)

For maths attainment at age 11 (KS2), a score of 4 is the national average, though most children will achieve a score between 3 and 5, with very few students achieving lower than this, and only very few, exceptional students attaining a score of 6. The mean score for maths attainment at age 11 within this study was 4.3 (s.d. = 0.7), in line with national standards.

Table 3 shows the model parameters for predictors of the intercept of maths attainment (i.e. at age 11). The substantive predictors of this study were parental mental health, parent–child interactions (mother and partner), gendered play (PSAI and CAI), home teaching (numeracy and literacy), parental education, parent–child relationships (harmony and control) and parental school involvement (age 8 and 11).

Of these variables, only parental education (O level, A level and degree), gendered play (at age 3.5; PSAI), school involvement at age 11, and a harmonious parent–child relationship were significantly associated with maths attainment at age 11. For parental education, when compared with children with parents with a CSE qualification or lower, greater maths attainment was predicted by children having parents with an O level ($b = 0.142$), an A level ($b = 0.219$) and a degree ($b = 0.378$). There was no statistically significant difference in attainment for children whose parents had a vocational qualification compared with a CSE qualification ($p = 0.597$). School involvement at age 11 (but not age 8; $p = 0.153$) equated to a significant increase in maths attainment by 0.022 levels per extra activity participated in. A harmonious parent–child relationship (but not child-control; $p = 0.209$), was associated with significantly higher maths attainment, with a 1-unit increase on the harmony scale equating to increases in maths by 0.038 levels.

Gendered play at age 3.5 was a significant predictor of maths attainment at age 11, where more masculine play and behaviours predicted decreased attainment. A 1-unit increase in PSAI score equated to decreased maths attainment by −0.002 levels. The remaining substantive predictors, namely parental

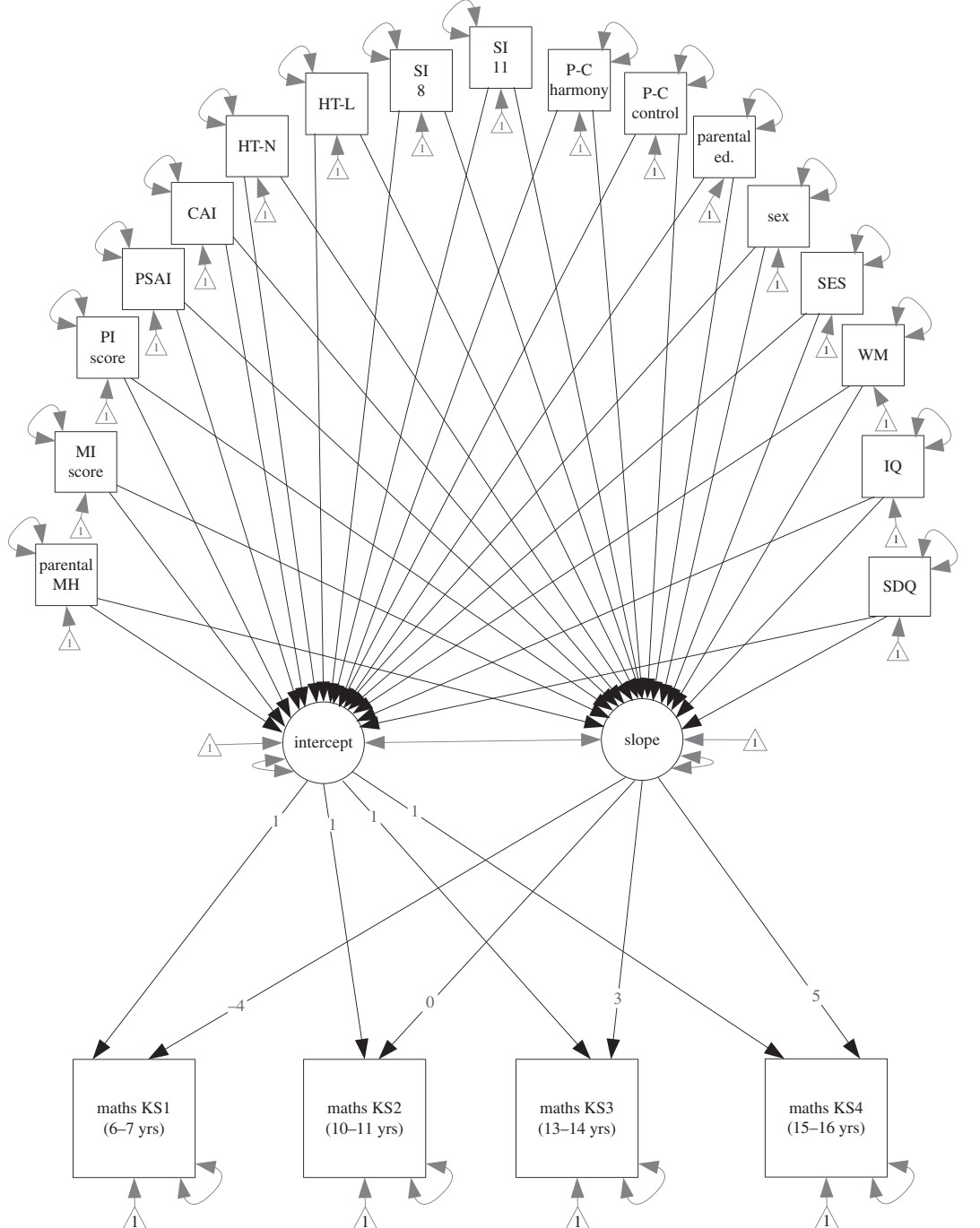

**Figure 1.** Latent growth model for maths attainment trajectories pre-transition to secondary education. The intercept represents maths attainment at age 11, and the slope represents maths attainment from age 7 to 16. Paths between predictor variables are implied but not illustrated. KS, key stage; MH, mental health; MI, mother's interaction score; PI, partner's interaction score, PSAI, gendered play (3.5 years); CAI, gendered play (8 years); HT-N, home teaching (numeracy); HT-L, home teaching (literacy); SI, school involvement; PC, parent–child, ed., education; WM, working memory; SDQ, internalizing symptoms.

mental health and aspects of the home environment (parent–child interactions, gendered play (age 8) and home teaching), did not significantly predict maths attainment at age 11 (table 3).

The contextual predictors included sex, internalizing symptoms, working memory, IQ and SES. A higher IQ, greater working memory, higher SES, and male sex were all significant predictors of greater maths attainment at age 11. A 10-unit increase in IQ equated to a 0.20 increase in maths attainment. For working memory, one additional trial completed correctly equated to an increase of 0.013 levels in maths attainment. A 10-unit increase in SES equated to a 0.04 increase in maths

**Table 2.** Correlation matrix for variables in the model predicting maths attainment.

| variable | M | s.d. | 1 | 2 | 3 | 4 | 5 | 6 | 7 | 8 | 9 | 10 | 11 | 12 | 13 | 14 | 15 | 16 | 17 | 18 | 19 |
|---|---|---|---|---|---|---|---|---|---|---|---|---|---|---|---|---|---|---|---|---|---|
| SDQ (1) | 2.39 | 2.55 | | −0.09 | −0.06 | −0.05 | 0.22 | −0.06 | −0.09 | −0.07 | −0.06 | 0.00 | 0.00 | 0.01 | −0.04 | 0.00 | −0.02 | −0.10 | −0.15 | −0.13 | −0.11 |
| IQ (2) | 106.44 | 15.50 | 0.00 | | 0.35 | 0.28 | −0.02 | 0.00 | 0.06 | 0.07 | 0.02 | 0.09 | 0.07 | 0.15 | 0.14 | 0.08 | 0.06 | 0.47 | 0.57 | 0.65 | 0.59 |
| WM (3) | 19.19 | 7.62 | 0.00 | 0.00 | | 0.17 | 0.00 | −0.04 | 0.00 | 0.02 | 0.00 | 0.04 | 0.05 | 0.08 | 0.08 | 0.03 | 0.06 | 0.26 | 0.35 | 0.36 | 0.33 |
| SES (4) | 59.17 | 11.69 | 0.00 | 0.00 | 0.00 | | −0.02 | 0.01 | 0.10 | 0.04 | −0.01 | 0.02 | 0.02 | 0.14 | 0.16 | 0.03 | 0.01 | 0.19 | 0.24 | 0.29 | 0.32 |
| parental MH (5) | −0.04 | 0.92 | 0.00 | 0.16 | 0.00 | 0.07 | | −0.06 | −0.10 | 0.00 | 0.00 | 0.01 | 0.01 | −0.05 | −0.08 | 0.00 | 0.01 | −0.04 | −0.04 | −0.05 | −0.06 |
| interaction (M) (6) | 23.36 | 3.25 | 0.00 | 0.73 | 0.02 | 0.39 | 0.00 | | 0.34 | −0.04 | −0.02 | 0.22 | 0.22 | 0.08 | 0.06 | 0.02 | 0.02 | 0.00 | −0.01 | 0.00 | 0.01 |
| interaction (P) (7) | 17.27 | 4.40 | 0.00 | 0.00 | 0.91 | 0.00 | 0.00 | 0.00 | | 0.37 | 0.05 | 0.15 | 0.15 | 0.05 | 0.10 | 0.04 | −0.01 | 0.05 | 0.05 | 0.08 | 0.08 |
| CAI (8) | 49.43 | 15.63 | 0.00 | 0.00 | 0.11 | 0.02 | 0.98 | 0.01 | 0.00 | | 0.60 | −0.04 | −0.03 | −0.01 | −0.05 | −0.02 | 0.08 | 0.09 | 0.09 | 0.08 | 0.02 |
| PSAI (9) | 49.16 | 15.51 | 0.00 | 0.12 | 0.99 | 0.46 | 0.76 | 0.13 | 0.00 | 0.00 | | −0.03 | −0.01 | 0.04 | −0.03 | −0.03 | 0.09 | 0.02 | 0.05 | 0.04 | 0.04 |
| HOME teaching (L) (10) | 0.99 | 0.74 | 0.92 | 0.00 | 0.00 | 0.17 | 0.25 | 0.00 | 0.00 | 0.00 | 0.02 | | 0.53 | 0.04 | 0.03 | 0.00 | 0.00 | 0.04 | 0.04 | 0.04 | 0.03 |
| home teaching (N) (11) | 1.33 | 0.74 | 0.80 | 0.00 | 0.00 | 0.25 | 0.44 | 0.00 | 0.00 | 0.05 | 0.43 | 0.00 | | 0.04 | 0.03 | 0.00 | −0.02 | 0.06 | 0.04 | 0.04 | 0.03 |
| SI (age 8) (12) | 1.85 | 1.09 | 0.74 | 0.00 | 0.00 | 0.00 | 0.01 | 0.00 | 0.01 | 0.52 | 0.05 | 0.01 | 0.40 | | 0.35 | 0.00 | 0.00 | 0.10 | 0.13 | 0.18 | 0.20 |
| SI (age 11) (13) | 1.75 | 1.07 | 0.01 | 0.00 | 0.00 | 0.00 | 0.00 | 0.00 | 0.00 | 0.01 | 0.06 | 0.05 | 0.03 | 0.00 | | 0.00 | −0.02 | 0.13 | 0.14 | 0.17 | 0.20 |
| parent–child (H) (14) | 3.24 | 0.80 | 0.74 | 0.00 | 0.05 | 0.07 | 0.90 | 0.16 | 0.01 | 0.20 | 0.04 | 0.81 | 0.35 | 0.00 | 0.65 | | −0.07 | 0.09 | 0.06 | 0.12 | 0.11 |
| parent–child (C) (15) | 1.95 | 1.13 | 0.31 | 0.00 | 0.00 | 0.46 | 0.35 | 0.09 | 0.00 | 0.00 | 0.00 | 0.97 | 0.15 | 0.00 | 0.68 | 0.00 | | 0.04 | 0.05 | 0.06 | 0.04 |
| KS1 Maths (16) | 2.29 | 0.54 | 0.00 | 0.00 | 0.00 | 0.00 | 0.01 | 0.71 | 0.48 | 0.00 | 0.24 | 0.01 | 0.06 | 0.00 | 0.00 | 0.00 | 0.03 | | 0.56 | 0.59 | 0.51 |
| KS2 Maths (17) | 4.33 | 0.70 | 0.00 | 0.00 | 0.00 | 0.00 | 0.00 | 0.58 | 0.00 | 0.00 | 0.00 | 0.01 | 0.04 | 0.00 | 0.00 | 0.00 | 0.00 | 0.00 | | 0.77 | 0.68 |
| KS3 Maths (18) | 6.25 | 1.17 | 0.00 | 0.00 | 0.00 | 0.00 | 0.00 | 0.82 | 0.00 | 0.00 | 0.11 | 0.01 | 0.01 | 0.00 | 0.00 | 0.00 | 0.00 | 0.00 | 0.00 | | 0.85 |
| KS4 Maths (19) | 7.37 | 1.57 | 0.00 | 0.00 | 0.00 | 0.00 | 0.00 | 0.70 | 0.00 | 0.12 | 0.21 | 0.00 | 0.03 | 0.00 | 0.00 | 0.00 | 0.01 | 0.00 | 0.00 | 0.00 | |

Note: The upper triangle displays the correlation coefficients and the lower triangle displays the $p$-values. SDQ, internalizing symptoms; WM, working memory; MH, mental health; interaction M and P, mother's home interaction and partner's home interaction; CAI, gendered play (8 years); PSAI, gendered play (3.5 years); home teaching N and L, numeracy and literacy; SI, school involvement; parent child H and C, harmony and control; KS, key stage.

**Table 3.** Model parameters for predictors of the intercept of maths attainment (at age 11).

| predictor | b | 95% CI | β | p-value |
|---|---|---|---|---|
| sex | 0.080 | [0.019, 0.142] | 0.055 | 0.011 |
| SDQ | −0.017 | [−0.024, −0.011] | −0.061 | 0.000 |
| IQ | 0.020 | [0.019, 0.022] | 0.433 | 0.000 |
| working memory | 0.013 | [0.010, 0.015] | 0.133 | 0.000 |
| SES | 0.004 | [0.002, 0.006] | 0.066 | 0.000 |
| Edu: CSE versus vocational | −0.024 | [−0.112, 0.064] | −0.007 | 0.597 |
| Edu: CSE versus O level | 0.142 | [0.078, 0.206] | 0.085 | 0.000 |
| Edu: CSE versus A level | 0.219 | [0.155, 0.283] | 0.143 | 0.000 |
| Edu: CSE versus degree | 0.378 | [0.304, 0.452] | 0.227 | 0.000 |
| parental mental health | −0.019 | [−0.037, 0.000] | −0.023 | 0.053 |
| mother's interaction score | −0.004 | [−0.009, 0.002] | −0.016 | 0.206 |
| partner's interaction score | 0.003 | [−0.001, 0.007] | 0.016 | 0.204 |
| CAI score | 0.001 | [−0.000, 0.002] | 0.021 | 0.169 |
| PSAI score | −0.002 | [−0.004, −0.000] | −0.041 | 0.048 |
| home teaching (literacy) | −0.001 | [−0.027, 0.026] | −0.001 | 0.960 |
| home teaching (numeracy) | 0.003 | [−0.023, 0.030] | 0.003 | 0.811 |
| school involvement (age 8) | 0.012 | [−0.005, 0.029] | 0.018 | 0.153 |
| school involvement (age 11) | 0.022 | [0.005, 0.039] | 0.032 | 0.012 |
| parent–child relationship (harmony) | 0.038 | [0.017, 0.059] | 0.042 | 0.000 |
| parent–child relationship (control) | 0.010 | [−0.005, 0.024] | 0.015 | 0.209 |

Note: β is the standardized parameter estimate.

attainment. Males' maths attainment was 0.080 levels higher than females'. Increased internalizing symptoms significantly predicted decreased maths attainment; a 1-unit increase in SDQ score equated to a decrease in maths attainment by −0.017 levels at age 11.

## 3.3. Predictors of the rate of change (maths attainment from age 7 to 16)

Table 4 shows the model parameters for predictors of the slope (i.e. the rate of change) of maths attainment from age 7 to 16. The group-level growth in maths attainment each year was 0.48, meaning that at average levels of the predictors, children progressed by close to half a national curriculum grade level each year, which is the expected progress in line with government recommendations.

Of the substantive variables specified above, parental education (O level, A level and degree), school involvement (at age 11), parental mental health, gendered play (at age 3.5; PSAI) and a harmonious parent–child relationship were significantly associated with the rate of change in maths attainment over time.

For parental education, when compared with children with parents with a CSE qualification, increased growth in maths attainment was predicted by children to parents with an O level ($b = 0.021$), an A level ($b = 0.048$) and a degree ($b = 0.086$). There was no statistically significant difference in attainment growth for children whose parents had a vocational qualification compared to a CSE qualification ($p = 0.929$). School involvement at age 11 (but not age 8; $p = 0.108$), equated to a significant increase in maths attainment growth by 0.004 levels per year for each extra activity participated in. A harmonious parent–child relationship (but not child-control; $p = 0.281$), was associated with a significantly faster rate of change in maths attainment, with a 1-unit increase on the harmony scale equating to increases in maths by 0.006 levels per year. Gendered play at age 3.5 significantly predicted a slower rate of change in maths attainment, with a 10-unit increase in PSAI score equating to a decrease in maths attainment per year by −0.004 levels. Parental mental health predicted a slower rate of change in maths attainment, with a 1-unit increase in symptoms equating to decreased maths attainment growth per year by −0.004

**Table 4.** Model parameters for predictors of the slope of maths attainment.

| predictor | b | 95% CI | β | p-value |
|---|---|---|---|---|
| sex | 0.014 | [0.002, 0.027] | 0.059 | 0.027 |
| SDQ | −0.002 | [−0.003, −0.001] | −0.044 | 0.003 |
| IQ | 0.003 | [0.002, 0.003] | 0.356 | 0.000 |
| working memory | 0.002 | [0.001, 0.002] | 0.111 | 0.000 |
| SES | 0.001 | [0.000, 0.001] | 0.077 | 0.000 |
| Edu: CSE versus vocational | −0.001 | [−0.018, 0.017] | −0.001 | 0.929 |
| Edu: CSE versus O level | 0.021 | [0.008, 0.033] | 0.076 | 0.001 |
| Edu: CSE versus A level | 0.048 | [0.035, 0.060] | 0.189 | 0.000 |
| Edu: CSE versus degree | 0.086 | [0.071, 0.101] | 0.314 | 0.000 |
| parental mental health | −0.004 | [−0.008, −0.000] | −0.031 | 0.036 |
| mother's interaction score | 0.000 | [−0.001, 0.001] | −0.003 | 0.826 |
| partner's interaction score | 0.000 | [−0.001, 0.001] | 0.008 | 0.620 |
| CAI score | 0.000 | [−0.000, 0.000] | −0.009 | 0.653 |
| PSAI score | 0.000 | [−0.001, −0.000] | −0.051 | 0.045 |
| home teaching (literacy) | −0.002 | [−0.007, 0.004] | −0.010 | 0.546 |
| home teaching (numeracy) | 0.002 | [−0.004, 0.007] | 0.010 | 0.559 |
| school involvement (age 8) | 0.003 | [−0.001, 0.006] | 0.025 | 0.108 |
| school involvement (age 11) | 0.004 | [0.001, 0.008] | 0.037 | 0.017 |
| parent–child relationship (harmony) | 0.006 | [0.001, 0.010] | 0.038 | 0.008 |
| parent–child relationship (control) | 0.002 | [−0.001, 0.005] | 0.016 | 0.281 |

Note: β is the standardized parameter estimate.

levels. Parent–child interactions, gendered play (at age 8; CAI) and home teaching, did not significantly predict the rate of change in maths attainment (table 4).

The significant contextual predictors of growth in maths attainment over time were sex (male), greater IQ, higher SES, greater working memory and fewer internalizing symptoms. Being male equated to increased growth of 0.014 levels per year in maths. A 10-unit increase in IQ and SES increased the rate of change by 0.03 and 0.01 levels, respectively. For each trial of the working memory task completed correctly, the rate of change in maths increased by 0.002. Greater internalizing symptoms decreased the rate of change in maths per year by −0.002 for a 1-unit increase in SDQ score. However, these effects are extremely small within the context of a group-level rate of change of 0.48 levels in maths per year.

# 4. Discussion

This study aimed to identify predictors of maths attainment trajectories across the transition from primary to secondary education, focusing on parental factors in childhood and adolescence. To recap, the study specifically investigated the following parental factors: mental health, home teaching, parent–child interactions at home, parent–child relationships, school involvement, education qualifications and child gendered play as substantive predictors of children's maths attainment trajectories.

## 4.1. Summary of results: parental predictors

Broadly, the results provided support for *some* of the parental factors as predictors of maths attainment at the transition (the intercept), and the growth over time (the slope). When looking at the intercept, the factors found to be significantly associated with higher maths attainment at age 11 were greater parental education, a harmonious parent–child relationship, greater school involvement at age 11 and 'feminine' gendered play at age 3.5 years. Whereas, when predicting the slope, significant effects were

found for greater parental education (increased rate of change (ROC)), poor parental mental health (slower ROC), 'masculine' gendered play at 3.5 years (slower ROC), school involvement at age 11 (increased ROC) and a harmonious parent–child relationship (increased ROC).

For parental education, the gains in attainment increased as parents' education qualifications increased (i.e. having parents with a degree equated to the highest attainment). Although, there were no significant differences in maths attainment at age 11 and the rate of change over time between children to parents with a CSE (and below), and children to parents with vocational qualifications. Unsurprisingly, parental education was found to be the strongest predictor of maths attainment at age 11, and of the rate in change over time, although, the effect sizes were slightly smaller in this study compared to previous analyses [62]. One of the aims of this study was to help identify the underlying mechanisms in which greater parental education contributes to higher maths attainment, i.e. through increased participation in home teaching, or in school activities for example. Given that aspects of parenting, the home environment, and contextual factors like SES were adjusted for in the model, it seems there are unique ways that parental education contributes to maths attainment. One explanation could be genetic [94,95]. For example, parents that are more highly educated may pass on traits that are important for educational attainment (such as motivation and temperament [96]), and are also more likely to provide an environment that is intellectually stimulating (this is referred to as the passive gene–environment correlation/passive rGE; [97,98]). Therefore, it could be that the differences between children of low- and highly educated parents may be based upon genetic (or gene × environment) factors which were not captured in this analysis. Indeed, genetic components involved in maths attainment have been identified by previous research (i.e. [99–101]); however, this explanation is out of the scope of this analysis which focused on parenting factors that could be actively (*and arguably easily*) changed by parents. However, this finding does provide future suggestions for further research within this area.

Greater school involvement at age 11, but not age 8, predicted greater maths attainment at the time of the transition (i.e. age 11), with an additional activity on the scale (between 0 and 4), equating to an increase in maths attainment by 0.022. Moreover, greater school involvement at age 11 significantly predicted an increased rate of change in maths attainment over time, however, as with the intercept, involvement at age 8 did not significantly predict growth in attainment. This finding is broadly in line with existing research where small effect sizes have been reported ($r = 0.13$; [16]), suggesting that greater parental interest and involvement in school activities is important for maths grades to some extent. Although, the minimal effect of involvement at age 8 was unexpected. As discussed previously, parental support across the transition can buffer against the negative effects associated with the transition [56–58], which could possibly explain why involvement at age 11 would be a stronger predictor than age 8 as it was measured just prior to the transition—meaning that those whose parents were more involved in school may have a more positive transition experience, thus reducing any negative effects on attainment.

A harmonious parent–child relationship at age 12 was associated with greater maths attainment at age 11 and an increased rate of change over time; whereas, child-control was not significantly associated with either. The finding that a harmonious parent–child relationship was linked to attainment was not surprising; however, the effect size is somewhat smaller than in the existing literature ($\beta = 0.12$–$0.18$; [102]). It was also expected that parent–child dyads where the child was more controlling (a proxy for autonomy) would be associated with greater attainment, as it has been previously found that children with parents that provide greater autonomy support have more positive social, emotional and academic outcomes (e.g. [61,103–106]). However, this idea was not supported by the findings of this study. It could be that 'child-control' was not a suitable proxy for child autonomy, or that neither parental control, nor child control is optimal for attainment, and that a more collaborative relationship is perhaps more beneficial for learning.

Gendered play at age 3.5 years, but not age 8, predicted the intercept and slope in maths attainment, with more 'masculine' play predicting lower maths attainment at age 11 and a slower rate of change over time. However, when placing these effects within the context of the scale of the PSAI (which ranges from 0 to 100), both effects are *extremely* small—a 10-unit increase in PSAI score would equate to a decrease of −0.02 in attainment at age 11, whereas, for the slope, even with a change of 100 units (i.e. the entire scale), the rate of change in maths attainment per year would be −0.04. It is important to note that the average rate of change per year is half a national curriculum level (i.e. 0.48), which illustrates the extremely minimal effects of gendered play found here.

Parental mental health within the first few months following the child's birth did not significantly predict maths attainment at age 11, but did marginally ($p = 0.05$) predict the slope in attainment in the

expected direction, where poorer mental health was linked to a slower rate of change. The effect of parental mental health on attainment was *extremely* small ($b = 0.019$), which was smaller than expected [13]. Previous research shows an increased risk of children not attaining a 'pass' grade in GCSE maths at age 16 whose mothers experience severe, persistent or recurrent postnatal depression ([63] OR = 2.65; [64] $\beta = 1.52$). These effects are much larger than what we found; however, in this study, we focused on parental mental health combined as an indicator of the home environment rather than solely investigating maternal effects. This difference may explain the inconsistent findings as poor paternal mental health has not been found to predict maths attainment in previous studies [64]. Additionally, as this variable was measured in the first few months of life, it means that any changes over time (i.e. throughout childhood and adolescence), were not accounted for. Additional research would be beneficial in assessing any association between the trajectory of parental mental health and child maths attainment across the transition.

Home interaction scores (mother and partner), and home teaching (numeracy and literacy), did not significantly predict maths attainment at age 11, nor the slope over time, which was unanticipated. Existing literature suggests that a greater participation in educational activities at home, and greater parent–child interaction is associated with higher maths (and general educational) attainment [24–28], although others have highlighted the inconsistencies within the literature [107]. The differences in findings of this study and the existing literature could be due to the measures used in this study. Home teaching was measured through dichotomous measures of 'yes' and 'no' which were then combined, meaning that the score for each (numeracy and literacy), could only range from 0 (all 'no') to 2 (all 'yes'). Moreover, parent–child home interactions were measured when the child was 3.5 and 6.75 years old and included various statements such as 'mother plays with child with toys' and 'mother reads stories with child'. It could be that the questions asked were too general, and included basic activities that a large proportion of parents would participate in with their child—meaning there would be little variation between participants, which is certainly true for the mother's interaction score. Whereas, the partner's interaction score had much more variance, though this was still not a significant predictor. It could be that the timepoints used were at a time where children are still heavily reliant upon their parents for participation in daily activities (such as feeding and bathing), and it may be that at a later age, where parental interaction is more varied between families, it is potentially a more appropriate timepoint to focus on when looking at differences between children based upon parental interactions.

In summary, it seems that parental education, school involvement (at age 11), and a harmonious parent–child relationship are the most important parent-related factors found in this study that predict maths attainment trajectories in adolescence. These findings could be explained somewhat by the stage-environment fit theory proposed by Eccles and colleagues [108,109]. The stage-environment fit theory describes that negative outcomes can occur when there is a mismatch between the developmental needs of an adolescent and the characteristics of their social environment. This means that adolescents whose social environments respond well to their changing needs are more likely to experience positive outcomes. It is possible that parents who are more involved in school are more aware of their child's changing needs across the transition to secondary education through greater interaction and discussion with their children (or their teachers), regarding school. Moreover, a positive (harmonious) parent–child relationship may mean that parents are perhaps more compassionate and accommodating when responding to their changing needs. It could be that when the changes in an adolescent's needs are appropriately met by their social environment, the transition to secondary education is more successful, and as such, is associated with increased maths attainment.

## 4.2. Summary of results: contextual predictors

Sex, SES and IQ predicted maths attainment at the intercept (age 11) and the slope, with being male, having higher SES and a higher IQ predictive of greater maths attainment and a quicker rate of change over time. As expected, internalizing symptoms and working memory predicted maths attainment trajectories, with a very small effect. These results are not discussed here in depth as they were included solely to adjust for them; see Evans *et al.* [62] for further discussion of these findings.

It was expected based upon the wider literature and a previous study conducted by the authors on the same dataset [62], that IQ and socio-economic status would predict maths attainment trajectories across the transition from primary to secondary education. Research investigating IQ and SES and attainment has found similar results [110–112], with this study adding further support to the existing literature.

In addition, it was expected that males would have greater maths attainment at age 11 [4], which was supported by the findings of this study. However, the analysis also showed that males had an increased rate of change over time, meaning that each year on average they made greater progress compared to their female peers, the effects were small, but this highlights the long-term negative effects associated with the gender gap in maths attainment.

## 4.3. Study limitations and future research directions

There are notable methodological issues that may affect the interpretation of these results. Firstly, the data were initially collected close to 30 years ago, meaning the findings could be less applicable now. This may be the case when investigating parental factors because the home environment has changed in the past 30 years with more mothers with young children in employment for example [113]. This possibility could mean that there are differences between maths attainment in this sample and children transitioning now, given that parental presence at home, for example, is linked to a successful education transition [59].

Additionally, children's social environment has changed in a number of ways since the participants in this sample transitioned to secondary education between the years of 2001 and 2004. These changes include the increase in adolescents owning mobile phones and using social media apps and sites [114,115], and increases in mental health issues [116], meaning that the effects of the transition may be somewhat different for students now. For example, the social media sites Facebook and Twitter were launched in 2004 and 2006, respectively, meaning few children in this study would have had access to these sites before transitioning, and it is unlikely that a large percentage of them would have used them during secondary education. Other popular apps such as Instagram and Snapchat were launched in 2010 and 2011 which would have been after this sample had finished secondary education entirely. Whereas, adolescents transitioning now are already likely to use many of these sites/apps before transitioning [117], or begin using them in early adolescence post-transition. Moreover, a report by the Children's Commissioner for England found that children using social media prior to secondary education focus on games and creative activities, whereas the focus post-transition is on 'likes' and 'comments', affecting their emotional well-being [117]. Increased social media and phone use in adolescence has been linked to heightened depression and suicidal ideation in adolescents by other researchers also [118]. These findings imply that a greater number of children transitioning now may encounter emotional difficulties around this period compared to children in this study, and given that emotional well-being predicts maths attainment trajectories [62], it is possible that transition experiences and attainment differs between these groups, which affects the generalizability of the findings. Additional research using more recent data would help to further understand the impact of the transition in light of these changes in children's environments, and how they potentially alter the impact of the transition on psychological and academic outcomes.

There are many advantages of using a large birth cohort such as ALSPAC, for example, the large sample size and breadth of topics assessed; however, there are also limitations including the high level of missing data, and the lack of depth for some of the measures. For example, in this study, numeracy home teaching was measured using parents' self-report of whether they had taught their child numbers and shapes, this does not account for the wide range of other maths and numeracy teaching activities (such as cooking together, handling money in shops, playing boardgames etc.) that help develop children's maths skills. Most of the measures also rely heavily on parents' abilities to identify their own behaviours and report them accurately and honestly. There are additional generalizability issues where children in ALSPAC achieve slightly higher grades in national curriculum exams at age 16 and are more likely to be white with higher socio-economic status compared to children not enrolled in the study [67], suggesting that it would be beneficial to conduct additional research with a more diverse sample.

The findings show that other than parental education, two of the most important factors found by this study are a harmonious parent–child relationship and parental school involvement at age 11. Although a positive parent–child relationship relies on numerous factors, parental school involvement can be increased more easily, hopefully leading to gains in maths attainment. However, the present study only looked at four different kinds of involvement with school, whereas there are many more ways parents can get involved with their child's education. For example, this study did not look at parent's help with homework, or their general interest in daily school life, which future research could focus on. Additional research investigating parental school involvement may help uncover which aspects are associated with the largest increases in maths, so that transition strategies could focus on improving these aspects.

## 4.4. Conclusion

The goal of the current study was to further understand which parental factors may influence maths attainment in adolescence. This study extends the existing literature by finding support for parental education qualifications, parental school involvement at age 11, and harmonious parent–child relationships as predictors of maths attainment. In addition, parental mental health in early childhood was not found to have a long-term impact and the findings show there is very little effect of gender-stereotyped play on maths attainment, suggesting that sex differences in maths attainment stem from other factors. General parent–child interactions and home teaching were also not found to predict maths attainment, suggesting that parents influence their child's maths attainment in other ways. However, due to some methodological limitations, additional research is still needed. Future exploration should aim to further uncover the relationship between parental education and children's maths attainment, with the goal to help close the associated gap in achievement between children to parents with higher qualifications and children to parents without educational qualifications. Failing to appropriately address these issues in early childhood further adds to the negative cycle of low maths attainment for parents and their children. Adults' low maths skills are associated with high unemployment rates and lower SES [6–8], which as found in this study, is linked to their child's maths attainment also. Therefore, further work is needed to eradicate the 'maths crisis' in the UK, thus improving several long-term outcomes for individuals and wider society.

Ethics. Ethical approval for this research was granted by the University of Sussex Cross-Schools Research Ethics Committee under submission code ER/DE84/1. Ethical approval was obtained from the ALSPAC Ethics and Law Committee and the Local Research Ethics Committees. Informed consent for the use of data collected via questionnaires and clinics was obtained from participants following the recommendations of the ALSPAC Ethics and Law Committee at the time.

Data accessibility. Data used for this submission will be made available on request to the Executive (alspacexec@bristol.ac.uk). The ALSPAC data management plan (http://www.bristol.ac.uk/alspac/researchers/data-access/documents/alspac-datamanagementplan.pdf) describes in detail the policy regarding data sharing, which is through a system of managed open access. Code for analysis is available at: https://osf.io/a5xsz/?view_only=87ae173f775b40d79d6cd0fdcf6d4a9c.

Authors' contributions. D.E. and A.P.F. conceived the study. D.E. conducted initial data processing and ran all statistical analyses. D.E. wrote the manuscript and A.P.F. reviewed and revised the manuscript and data analysis process at all stages. All authors gave final approval for publication.

Competing interests. We declare we have no competing interests.

Funding. The UK Medical Research Council and the Wellcome Trust (Grant ref: 102215/2/13/2) and the University of Bristol provide core support for ALSPAC. A comprehensive list of grants funding is available on the ALSPAC website (http://www.bristol.ac.uk/alspac/external/documents/grant-acknowledgements.pdf). This research was specifically funded by Department for Education and Skills (Grant ref: EOR/SBU/2002/121) and the Wellcome Trust and MRC (Grant ref: 092731). This publication is the work of the authors and they will serve as guarantors for the contents of this paper. This specific research project did not receive any funding.

Acknowledgements. The authors are extremely grateful to all the families who took part in this study, the midwives for their help in recruiting them, and the whole ALSPAC team, which includes interviewers, computer and laboratory technicians, clerical workers, research scientists, volunteers, managers, receptionists and nurses.

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
