## [Reviewer comments · Royal Society Open Science]

Review History

RSOS-200422.R0 (Original submission)

Review form: Reviewer 1

Is the manuscript scientifically sound in its present form?

Yes

Are the interpretations and conclusions justified by the results?

Yes

Is the language acceptable?

Yes

Do you have any ethical concerns with this paper?

No

Have you any concerns about statistical analyses in this paper?

No

Recommendation?

Accept with minor revision (please list in comments)

Comments to the Author(s)

The manuscript reviews current literature well and has been well thought out. The introduction is informative and well structured. Method is also very informative with great level of detail and nice breakdown of statistical analysis. Results are structured nicely and easy to follow. In the discussion results are nicely explained and backed up with relevant literature. Minor changes to be made as follows:

Introduction

- Page 3 line 36 and line 38 both start with 'parents can also' and 'parents also', change wording so not repetitive.

Method

Sample

- The following information is confusing- is this two samples (i.e. a core sample and additional sample)? Could this be made clearer for the reader?
- "The core sample consisted of 14,062 live births, of which 13,988 children were alive at 1 year. Additional participants were recruited resulting in a total of 15,589 fetuses, of which 14,901 were alive at 1 year."

(iv) Parental education

- Justify why CSE (or below) was used as a reference group.

Results

- Table 1 and 2 Cannot see top of table clearly to see what the numbers represent. May be an issue with journal format rather than your tables?
- Table 2. Could you provide the numbers of the variables beside the variables names so that the reader does not have to count along the variables?
- Table 2. Providing the coefficient and p value significance marked as e.g. 0.28** for .001 significance might be easier for the reader as there is a lot of searching to be done to find the p value for each coefficient. This would make it a lot quicker for the reader.
- Table 3. Should there be a reference within table 3 that this is predictors of intercepts of maths attainment at age 11?

Discussion

- Page 16, line 47 needs a lower-case letter change with the word "However, When".
- Could the authors provide some literature on gendered play to explain their findings?
- Could the authors add more information regarding what is normally seen in literature regarding parent mental health throughout childhood and adolescence and child outcomes as a summary statement of this paragraph (page 17, line 3).

Great paper, well done.

Review form: Reviewer 2

Is the manuscript scientifically sound in its present form?

Yes

Are the interpretations and conclusions justified by the results?

No

Is the language acceptable?

Yes

Do you have any ethical concerns with this paper?

No

Have you any concerns about statistical analyses in this paper?

Yes

Recommendation?

Major revision is needed (please make suggestions in comments)

Comments to the Author(s)

All of my comments to the author(s) are in the attached file (Appendix A).

Decision letter (RSOS-200422.R0)

20-May-2020

Dear Ms Evans

On behalf of the Editors, I am pleased to inform you that your Manuscript RSOS-200422 entitled "Predictors of Mathematical Attainment Trajectories across the Primary-to-Secondary Education Transition: Parental Factors and the Home Environment" has been accepted for publication in Royal Society Open Science subject to minor revision in accordance with the referee suggestions. Please find the referees' comments at the end of this email.

The reviewers and handling editors have recommended publication, but also suggest some minor revisions to your manuscript. Therefore, I invite you to respond to the comments and revise your manuscript.

- Ethics statement

- Data accessibility

<http://datadryad.org/submit?journalID=RSOS&manu=RSOS-200422>

- Competing interests

- Authors' contributions

- Acknowledgements

- Funding statement

Because the schedule for publication is very tight, it is a condition of publication that you submit the revised version of your manuscript before 29-May-2020. Please note that the revision deadline will expire at 00.00am on this date. If you do not think you will be able to meet this date please let me know immediately.

- 1) A text file of the manuscript (tex, txt, rtf, docx or doc), references, tables (including captions) and figure captions. Do not upload a PDF as your "Main Document";

- 2) A separate electronic file of each figure (EPS or print-quality PDF preferred (either format should be produced directly from original creation package), or original software format);
- 3) Included a 100 word media summary of your paper when requested at submission. Please ensure you have entered correct contact details (email, institution and telephone) in your user account;
- 4) Included the raw data to support the claims made in your paper. You can either include your data as electronic supplementary material or upload to a repository and include the relevant doi within your manuscript. Make sure it is clear in your data accessibility statement how the data can be accessed;
- 5) All supplementary materials accompanying an accepted article will be treated as in their final form. Note that the Royal Society will neither edit nor typeset supplementary material and it will be hosted as provided. Please ensure that the supplementary material includes the paper details where possible (authors, article title, journal name).

If your manuscript is newly submitted and subsequently accepted for publication, you will be asked to pay the article processing charge, unless you request a waiver and this is approved by Royal Society Publishing. You can find out more about the charges at <https://royalsocietypublishing.org/rsos/charges>. Should you have any queries, please contact openscience@royalsociety.org.

on behalf of Dr Emma Hayiou-Thomas (Associate Editor)
openscience@royalsociety.org

Associate Editor Comments to Author (Dr Emma Hayiou-Thomas):

Comments to the Author:

I enjoyed seeing the next step in your work on maths attainment in the ALSPAC sample, and the interesting findings in relation to the home environment. As both reviewers commented, the paper is clearly written, and has many strengths. There are some non-trivial changes which need to be made to the manuscript, but they are largely additional explanations, rather than re-analysis or major reframing. Please address each of the reviewers' constructive comments and suggestions - I would particularly like to highlight the following issues:

- A clearer motivation in the introduction for including gendered play as a potential predictor of maths attainment (R.1).
- Previous relevant findings on this topic from ALSPAC (including your own previous work!) (R.2).
- Exclusion criteria and potential for alpha slippage (R. 2).

In addition, I would also like to see a (brief) discussion on the pros and cons of using a large dataset such as ALSPAC - it clearly has major advantages in terms of the size of the dataset (N and breadth) but also inherent limitations in terms of depth of measurement, attrition, and cohort effects. I think this would provide helpful context. Finally, the discussion would benefit from an explicit consideration of how the effect sizes you report for your predictors compare to those previously reported in the literature for those predictors, and how your study extends the existing knowledge base.

Reviewer comments to Author:

Reviewer: 1

Comments to the Author(s)

The manuscript reviews current literature well and has been well thought out. The introduction is informative and well structured. Method is also very informative with great level of detail and nice breakdown of statistical analysis. Results are structured nicely and easy to follow. In the discussion results are nicely explained and backed up with relevant literature. Minor changes to be made as follows:

Introduction

- Page 3 line 36 and line 38 both start with 'parents can also' and 'parents also', change wording so not repetitive.

Method

Sample

- The following information is confusing- is this two samples (i.e. a core sample and additional sample)? Could this be made clearer for the reader?
- "The core sample consisted of 14,062 live births, of which 13,988 children were alive at 1 year. Additional participants were recruited resulting in a total of 15,589 fetuses, of which 14,901 were alive at 1 year."

(iv) Parental education

- Justify why CSE (or below) was used as a reference group.

Results

- Table 1 and 2 Cannot see top of table clearly to see what the numbers represent. May be an issue with journal format rather than your tables?
- Table 2. Could you provide the numbers of the variables beside the variables names so that the reader does not have to count along the variables?
- Table 2. Providing the coefficient and p value significance marked as e.g. 0.28** for .0.01 significance might be easier for the reader as there is a lot of searching to be done to find the p value for each coefficient. This would make it a lot quicker for the reader.
- Table 3. Should there be a reference within table 3 that this is predictors of intercepts of maths attainment at age 11?

Discussion

- Page 16, line 47 needs a lower-case letter change with the word "However, When".
- Could the authors provide some literature on gendered play to explain their findings?

- Could the authors add more information regarding what is normally seen in literature regarding parent mental health throughout childhood and adolescence and child outcomes as a summary statement of this paragraph (page 17, line 3).

Great paper, well done.

Reviewer: 2

Comments to the Author(s)

All of my comments to the author(s) are in the attached file.

Author's Response to Decision Letter for (RSOS-200422.R0)

See Appendix B.

Decision letter (RSOS-200422.R1)

13-Jun-2020

Dear Ms Evans,

It is a pleasure to accept your manuscript entitled "Predictors of Mathematical Attainment Trajectories across the Primary-to-Secondary Education Transition: Parental Factors and the Home Environment" in its current form for publication in Royal Society Open Science.

Please ensure that you send to the editorial office individual files for table included in your manuscript. You can send these in a zip folder if more convenient. Failure to provide these files may delay the processing of your proof.

Kind regards,

Andrew Dunn

on behalf of Dr Emma Hayiou-Thomas (Associate Editor)
openscience@royalsociety.org

Appendix A

Thank you for the opportunity to review the article “Predictors of Mathematical Attainment Trajectories across the Primary-to-Secondary Education Transition: Parental Factors and the Home Environment,” (RSOS-200422) which was submitted for consideration of publication in the Royal Society Open Science. The study is organized, well-written and describes the relation among key variables related to mathematics development throughout life (using the ALSPAC database), as the participants in the sample are now approximately 28 years old. I especially enjoyed the discussion of the early developmental trajectories in relation to later academic achievement, and the reference to Eccles work. Literature review appeared to be complete and succinct in relation to the data studied. This review details some concerns I have about the study and these concerns should be addressed before a decision about publication can be made. They are detailed below in order of importance.

(1) The authors admit that these data are based on a cohort that is now almost 30 years old. This discussion needs to occur in the Introduction, not as an afterthought in the Discussion section as a limitation. For example, there are differences in exposure to technology (no cell phones, tablets, internet), curriculum and inclusion policies (to name a few) that could affect children’s learning. It would be interesting to speculate how this cohort was influenced by societal influences at the time they were learning math. Further, there could be some detailing of the kinds of research that has already been conducted on this cohort (i.e., what is already known about these participants), and how this current study relates to previous work on the cohort.

(2) According to my search engine, over 1800 articles have been published using the ALSPAC database. The data set includes records on over 14,000 participants and their families and schools. What measures have been taken to protect against alpha slippage in this current study? Analyses were conducted on data that ranged from 3429 to 7263 participants, depending on the variable selected. With the size of this sample, it would be relatively easy to obtain significance by including various variables in question, and given very basic a priori hypotheses, many relations could be deemed significant.

(3) Related to sample size, I am concerned that “2652 students were excluded because they had special needs” as determined by their teachers and schools. Almost 20% of the sample was excluded (which goes well beyond the 10% of children estimated to have learning concerns in current populations). Too many children were excluded for reasons that are unclear. Additionally, more participants were excluded because of missing data, resulting in over half of the sample not being included in most analyses. Although it is powerful to study large cohorts of the population, I question the information that is not being detected as a result of missing or ignored data.

(4) The variables used in the study are quite simplified compared to math measures used in current day research. For example, instead of using math measures as an outcome variable, school grades were used instead (which incur obvious bias with teacher ratings and school context). Home learning variables were based on gender-type play and general parent involvement. Homework help was included in a long list of other parent involvement variables,

where today, this variable is studied on its own. Color knowledge was grouped with literacy proficiency and shape knowledge was considered with numeracy proficiency, for no apparent reason, and based on parent ratings. I would complete factor analyses on some of these large composite variables to try and add meaning (e.g., early home teaching, gender stereotyped play, and home interactions).

(5) “Parent education,” “parent school involvement,” “harmonized learning” and “female gendered play” relating to mathematics attainment at age 11. However, home interactions (at age 3 and 7), parental mental health, early home teaching, working memory and intelligence were not found to be related. All of these variables have been found to relate to children’s math development in various studies. Because of the enormity of the study, and the inability to delve into some of these measures, the overall picture is difficult to interpret. I wonder why some variables were significant and others were not.

Minor concerns:

- (1) How many observations were completed for the parent-child relationship and the working memory and intelligence variables? How were scores computed? It is unclear from the Tables.
- (2) It is difficult to read the Table headers. What is %MD? I assume Missing Data- if so, why are these numbers so high? What is KSI? Please include scales for all of your variables (e.g., working memory and intelligence).
- (3) The correlation table is overwhelming. Are all of these variables necessary? What story are you trying to tell? Which correlations meet significance, and at what alpha level? What are your effect sizes?
- (4) Why was masculinized gender play associated with decreased math attainment, and boys outperformed girls with their math grades at age 11? Is there a teacher bias going on here?
- (5) Please explain your rate of change analyses and provide a rationale for the purpose of these analyses in line with the hypotheses of the study.

Appendix B

Author response to reviews of

Manuscript RSOS-200422

Predictors of Mathematical Attainment Trajectories across the Primary-to-Secondary Education Transition: Parental Factors and the Home Environment

submitted to *Royal Society Open Science*

RC: Reviewer Comment AR: Author Response Manuscript text

Dear Dr Hayiou-Thomas,

Thank you very much for taking the time to consider our manuscript for publication at *Royal Society Open Science*. In the following we address your own and the reviewers' concerns and suggestions, and describe the revisions made to the manuscript in light of these.

1. Reviewer #1

RC: Page 3 line 36 and line 38 both start with 'parents can also' and 'parents also', change wording so not repetitive.

AR: Line 36 has been changed to:

There is also the opportunity for positive experiences for growth and development in childhood that are provided by parents.

RC: The following information is confusing - is this two samples (i.e. a core sample and additional sample)? Could this be made clearer for the reader? "The core sample consisted of 14,062 live births, of which 13,988 children were alive at 1 year. Additional participants were recruited resulting in a total of 15,589 fetuses, of which 14,901 were alive at 1 year."

AR: The core sample refers to the starting sample recruited by ALSPAC i.e., before any participant withdrawal. This has been changed to:

The core ALSPAC sample recruited initially consisted of 14,062 live births, of which 13,988 children were alive at 1 year. Additional participants were recruited resulting in a total of 15,589 fetuses, of which 14,901 were alive at 1 year.

RC: Justify why CSE (or below) was used as a reference group.

AR: We used CSE (or below) as the reference group as this was the lowest level of education available - this point has been made clearer in the method section quoted below.

Having the highest qualification of a CSE (or below) was used as the reference group as this was the lowest level of parental education qualifications available.

RC: Table 1 and 2 Cannot see top of table clearly to see what the numbers represent. May be an issue with journal format rather than your tables?

AR: Apologies they are unclear - it is an issue with the Latex class used.

RC: Table 2. Could you provide the numbers of the variables beside the variables names so that the reader does not have to count along the variables?

AR: Numbers have been added to the variables.

RC: Table 2. Providing the coefficient and p value significance marked as e.g. 0.28 for .001 significance might be easier for the reader as there is a lot of searching to be done to find the p value for each coefficient. This would make it a lot quicker for the reader.**

AR: A high number of the correlations are significant due to the very large sample size and we felt it was somewhat misleading to provide marked significance for different alpha levels due to the high proportion of significant p-values where many of the correlation coefficients (i.e., the effect sizes) are very small.

RC: Table 3. Should there be a reference within table 3 that this is predictors of intercepts of maths attainment at age 11?

AR: Added to the caption that this refers to age 11.

RC: Page 16, line 47 needs a lower-case letter change with the word “However, When”.

AR: Thank you for highlighting - this has been corrected.

RC: Could the authors provide some literature on gendered play to explain their findings?

AR: Following this recommendation and that of Dr Hayiou-Thomas, we have included the below in the introduction to highlight why gendered play is included in the analysis, but felt the effect was too trivial to discuss further than what is already in the discussion.

Parents additionally guide their children’s interests and the types of play they participate in by purchasing toys and encouraging (or discouraging) different types of activities and behaviours. One specific area of interest particularly for maths attainment is gender-stereotyped play in childhood. Typically, ‘boy toys’ include construction toys (such as building blocks and tools), vehicles and sports, and ‘girl toys’ include dolls, household toys (i.e., tea sets and toy kitchens), and ‘dress-up’ [38]. This divide is particularly interesting given reported differences in maths attainment between males and females [4], which could potentially stem from the differences in toys and play through the increased spatial content in ‘masculine’ toys for example. Because parents heavily shape their child’s preferences and activities, examining gender-stereotyped play in childhood could help understand differences in attainment for males and females stemming from parents and aspects of the home environment.

RC: Could the authors add more information regarding what is normally seen in literature regarding parent mental health throughout childhood and adolescence and child outcomes as a summary statement of this paragraph (page 17, line 3).

AR: This section has been edited to the below:

Parental mental health within the first few months following the child's birth did not significantly predict maths attainment at age 11, but did marginally ($p = .05$) predict the slope in attainment in the expected direction where poorer mental health was linked to a slower rate of change. The effect of parental mental health on attainment was extremely small ($b = 0.019$) 16 which was smaller than expected [15]. Previous research shows an increased risk of children not attaining a 'pass' grade in GCSE maths at age 16 whose mothers experience severe, persistent, or recurrent postnatal depression [63, OR = 2.65; 64, beta = 1.52]. These effects are much larger than what we found, however, in this study we focused on parental mental health combined as an indicator of the home environment rather than solely investigating maternal effects. This difference may explain the inconsistent findings as poor paternal mental health has not been found to predict maths attainment in previous studies [64]. Additionally, as this variable was measured in the first few months of life, it means that any changes over time (i.e., throughout childhood and adolescence), were not accounted for. Additional research would be beneficial in assessing any association between the trajectory of parental mental health and child maths attainment across the transition.

2. Reviewer #2

RC: The authors admit that these data are based on a cohort that is now almost 30 years old. This discussion needs to occur in the Introduction, not as an afterthought in the Discussion section as a limitation. For example, there are differences in exposure to technology (no cell phones, tablets, internet), curriculum and inclusion policies (to name a few) that could affect children's learning. It would be interesting to speculate how this cohort was influenced by societal influences at the time they were learning math. Further, there could be some detailing of the kinds of research that has already been conducted on this cohort (i.e., what is already known about these participants), and how this current study relates to previous work on the cohort.

AR: Societal differences between the cohort and children now have been added into the discussion (first quote), and a discussion of the pros and cons of using ALSPAC has been added also (quote 2). More information on the findings from studies using the ALSPAC cohort has been added to the introduction (quote 3).

Additionally, children's social environment has changed in a number of ways since the participants in this sample transitioned to secondary education between the years of 2001 and 2004. These changes include the increase in adolescents owning mobile phones and using social media apps and sites [114, 115], and increases in mental health issues [116] meaning that the effects of the transition may be somewhat different for students now. For example, the social media sites Facebook and Twitter were launched in 2004 and 2006 respectively, meaning few children in this study would have had access to these sites before transitioning, and it is unlikely that a large percentage of them would have used them during secondary education. Other popular apps such as Instagram and Snapchat were launched in 2010 and 2011 which would have been after this sample had finished secondary education entirely. Whereas, adolescents transitioning now are already likely to use many of these sites/apps before transitioning [117], or begin using them in early adolescence post-transition. Moreover, a report by the Children's Commissioner for England has found that children using social media prior to secondary education focus on games and creative activities, whereas the focus post transition is on "likes" and "comments", affecting their emotional wellbeing [117]. Increased social media and phone use in adolescence has been linked to heightened depression and suicidal ideation in adolescents by other researchers also [118]. These findings imply that a greater number of children transitioning now may encounter emotional difficulties around this period compared to children in this study, and given that emotional wellbeing predicts maths attainment trajectories [62], it is possible that transition experiences and attainment differs between these groups which affects the generalisability of the findings. Additional research utilising more recent data would help to further understand the impact of the transition in light of these changes in children's environments, and how they potentially alter the impact of the transition on psychological and academic outcomes.

There are many advantages of using a large birth cohort such as ALSPAC, for example the large sample size and breadth of topics assessed, however, there are also limitations including the high level of missing data, and the lack of depth for some of the measures. For example, in this study, numeracy home-teaching was measured using parents' self-report of whether they had taught their child numbers and shapes, this does not account for the wide range of other maths and numeracy teaching activities (such as cooking together, handling money in shops, playing boardgames etc.) that help develop children's maths skills. Most of the measures also rely heavily on parents' abilities to identify their own behaviours and report them accurately and honestly. There are additional generalisability issues where children in ALSPAC achieve slightly higher grades in national curriculum exams at age 16 and are more likely to be white with higher socio-economic status compared to children not enrolled in the study [67] suggesting that it would be beneficial to conduct additional research with a more diverse sample.

ALSPAC is a large UK birth cohort following children and their parents from pregnancy up to the present day, and covers a broad range of measures. Previous work investigating home and parental factors using the ALSPAC cohort have focused particularly on the impact of mothers and have shown that that maths attainment is associated with maternal perinatal and postnatal mental health [63,64], and maternal prenatal locus of control [65]. Parental education qualifications at birth are also linked to maths attainment in the ALSPAC sample [62], though, mothers' participation in adult learning was not found to improve maths grades [66]. The current study aims to add to this existing literature by focusing on the influence of the home environment and parental factors on the trajectory of children's maths attainment (measured from age 7 up to age 16 using national curriculum assessments).

RC: According to my search engine, over 1800 articles have been published using the ALSPAC database. The data set includes records on over 14,000 participants and their families and schools. What measures have been taken to protect against alpha slippage in this current study? Analyses were conducted on data that ranged from 3429 to 7263 participants, depending on the variable selected. With the size of this sample, it would be relatively easy to obtain significance by including various variables in question, and given very basic a priori hypotheses, many relations could be deemed significant.

AR: Rightfully said, many of the effects could be statistically significant resulting from the large sample size, to avoid being misleading we tried to focus on discussing the actual size of the effect in the context of the scales used, the average change in attainment per year, and whether the effect is meaningful/has practical applications rather than to focus on the specific p-values. An example of this is given below where the effect of gendered play was significant but the size of the effect in practical terms was essentially zero.

Gendered play at age 3.5 years, but not age 8, predicted the intercept and slope in maths attainment, with more “masculine” play predicting lower maths attainment at age 11 and a slower rate of change over time. However, When placing these effects within the context of the scale of the PSAI (which ranges from 0-100), both effects are extremely small - a 10-unit increase in PSAI score would equate to a decrease of -0.02 in attainment at age 11, whereas, for the slope, even with a change of 100 units (i.e., the entire scale), the rate of change in maths attainment per year would be -0.04. It is important to note that the average rate of change per year is half a national curriculum level (i.e., 0.48), which illustrates the extremely minimal effects of gendered play found here.

RC: Related to sample size, I am concerned that “2652 students were excluded because they had special needs” as determined by their teachers and schools. Almost 20% of the sample was excluded (which goes well beyond the 10% of children estimated to have learning concerns in current populations). Too many children were excluded for reasons that are unclear. Additionally, more participants were excluded because of missing data, resulting in over half of the sample not being included in most analyses. Although it is powerful to study large cohorts of the population, I question the information that is not being detected as a result of missing or ignored data.

AR: Children with SEN were excluded due to the anticipated effects this would have on their attainment. This group includes children identified as having (or have had) the following:

- Learning difficulties
- Specific learning difficulties (e.g. Dyslexia)
- Emotional and behavioural difficulties
- Speech and language difficulties
- Sensory impairment (Hearing)
- Sensory impairment (Visual)
- Physical disabilities
- Medical conditions
- Developmental delay
- Other

The breadth of special educational needs in the above criterion explains the high number of exclusions compared to what is seen in the population for general learning difficulties, and we felt the exclusions were justified because the model would not be useful (i.e., would not have good fit) in predicting maths attainment

for individuals with SEN (or for typically developing children) due to the high heterogeneity within this group. We did not have access to the above SEN data for individual children within the sample that we obtained from ALSPAC, so could not look at this group separately taking their specific SEN into consideration. To increase clarity, we have added that this study focuses on typically developing children into the introduction and added additional information to the method section as per the below:

Therefore, by utilising secondary analysis of the Avon Longitudinal Study of Parents and Children (ALSPAC), this study aims to investigate parental influences in childhood and early adolescence as predictors of maths attainment trajectories for typically-developing children across the transition from primary to secondary education.

2,652 children identified by teachers (at ages 7-8 and 10-11) as having or have had special educational needs (such as learning difficulties, emotional and behavioural difficulties, physical disabilities and speech and language difficulties) were excluded ($N = 11,832$) due to the high heterogeneity within this group.

In terms of the large amount of missing data, a significant proportion (37%) of the final sample were excluded as they had over 50% of their responses missing. We deemed it necessary to exclude participants where the level of missing data was over 50% because we conducted multiple imputation in the growth model and we felt that it would be inaccurate to impute data based on a sample where so many responses are missing.

- RC:** The variables used in the study are quite simplified compared to math measures used in current day research. For example, instead of using math measures as an outcome variable, school grades were used instead (which incur obvious bias with teacher ratings and school context). Home learning variables were based on gender-type play and general parent involvement. Homework help was included in a long list of other parent involvement variables, where today, this variable is studied on its own. Color knowledge was grouped with literacy proficiency and shape knowledge was considered with numeracy proficiency, for no apparent reason, and based on parent ratings. I would complete factor analyses on some of these large composite variables to try and add meaning (e.g., early home teaching, gender stereotyped play, and home interactions).
- AR: Factor analysis was conducted to create the composites which is detailed in the code on the OSF repository, but this information was left out of the manuscript which has now been added into the method section.
- RC:** “Parent education,” “parent school involvement,” “harmonized learning” and “female gendered play” relating to mathematics attainment at age 11. However, home interactions (at age 3 and 7), parental mental health, early home teaching, working memory and intelligence were not found to be related. All of these variables have been found to relate to children’s math development in various studies. Because of the enormity of the study, and the inability to delve into some of these measures, the overall picture is difficult to interpret. I wonder why some variables were significant and others were not.
- AR: We felt this is covered in the discussion.
- RC:** How many observations were completed for the parent-child relationship and the working memory and intelligence variables? How were scores computed? It is unclear from the Tables.
- AR: The above measures were assessed at a single timepoint. We felt this information was clear from the method section, but have added more information on the working memory measure in the method.

RC: It is difficult to read the Table headers. What is %MD? I assume Missing Data- if so, why are these numbers so high? What is KSI? Please include scales for all of your variables (e.g., working memory and intelligence).

AR: This is an issue with the Latex class used, the definition of MD was given in the caption but we have edited this to make it clearer. KSI refers to key stages (which are the assessments used in schooling in the UK) - this abbreviation is now defined in the caption.

RC: The correlation table is overwhelming. Are all of these variables necessary? What story are you trying to tell? Which correlations meet significance, and at what alpha level? What are your effect sizes?

AR: The matrix includes the raw correlations so readers are able to see the relationship between variables in the growth model, and theoretically they could reconstruct the analysis from the correlation matrix. So, all the variables *are* necessary.

Which correlations meet significance? The table includes the *p*-values for all correlations so it is clear which ones are significant.

What are your effect sizes? The correlation coefficient is an effect size.

In summary, the previous version of the table contained all of the information the reviewer needed to answer questions about effect size and significance. Furthermore, in a sample size this large we feel it is important not to draw undue attention to significance by using ***, it is better that readers process the correlation coefficients themselves because many small effects are, in fact, significant in this sample.

RC: Why was masculinized gender play associated with decreased math attainment, and boys outperformed girls with their math grades at age 11? Is there a teacher bias going on here?

AR: Due to the extremely small, and trivial effect of gendered play (betas = -0.002 & -0.0004), we felt a further discussion of this effect was not appropriate as the significant *p*-value is likely due to the large sample size. We tried to highlight this within the discussion by focusing on the size of the effect rather than the significance of the *p*-value as stated below:

Gendered play at age 3.5 years, but not age 8, predicted the intercept and slope in maths attainment, with more “masculine” play predicting lower maths attainment at age 11 and a slower rate of change over time. However, When placing these effects within the context of the scale of the PSAI (which ranges from 0-100), both effects are extremely small - a 10-unit increase in PSAI score would equate to a decrease of -0.02 in attainment at age 11, whereas, for the slope, even with a change of 100 units (i.e., the entire scale), the rate of change in maths attainment per year would be -0.04. It is important to note that the average rate of change per year is half a national curriculum level (i.e., 0.48), which illustrates the extremely minimal effects of gendered play found here.

RC: Please explain your rate of change analyses and provide a rationale for the purpose of these analyses in line with the hypotheses of the study.

AR: We used a latent growth model predicting maths attainment to measure attainment at age 11 and the growth in attainment over time. Anyone familiar with growth models would know that the slope represents the rate of change. However, in the description of the statistical analysis we say this explicitly:

The predictors were included as exogenous observed variables that predict the intercept and slope (i.e., the rate of change) of growth in maths attainment.

We have also included a figure of the model for clarity.

We are extremely appreciative of your reviews especially under the current circumstances and hope that these amendments are responsive to your recommendations.

Yours sincerely

Danielle Evans